# What Drives the Value of a *Shigella* Vaccine?

**DOI:** 10.3390/vaccines10020282

**Published:** 2022-02-12

**Authors:** William P. Hausdorff, Suzanne Scheele, Birgitte K. Giersing

**Affiliations:** 1Center for Vaccine Innovation and Access, PATH, 455 Massachusetts Ave NW, Washington, DC 20001, USA; sscheele@path.org; 2Faculty of Medicine, Université Libre de Bruxelles, 1050 Brussels, Belgium; 3Immunization, Vaccines and Biologicals Department, World Health Organization, 1211 Geneva, Switzerland; giersingb@who.int

**Keywords:** value proposition, investment case, full value of vaccines assessment

## Abstract

The development and licensure of a safe and highly efficacious *Shigella* vaccine has been a priority in international public health circles for decades and would represent a great scientific achievement. Nonetheless, in the context of increasingly crowded and costly childhood immunization programs, and with a myriad of other new and improved vaccines currently or soon on the market, there is no guarantee that even a highly effective *Shigella* vaccine would become a priority for adoption and introduction by the low- and middle-income countries that could benefit from it the most. We discuss here some of the major determinants and questions regarding the introduction of *Shigella* vaccines and the importance of developing a succinct, compelling public health value proposition.

## 1. The Need to Articulate the Value of a Vaccine: General Considerations

Proof of concept was established more than a decade ago that a vaccine effective in adults against the major diarrheal disease pathogen *Shigella* [1] could also prevent the disease in children [2]. Based on those studies and the multitude of Shigella vaccine candidates in Phase II studies, it is easy to be optimistic that at least one vaccine against *Shigella* will be licensed within the next 5–10 years with the aspirational attributes described in the World Health Organization’s (WHO) Preferred Product Characteristics (PPC) profile [3]. The development of such vaccines, long a priority of the WHO and other global organizations, would constitute a major scientific advance offering significant public health benefits. Yet it remains quite possible that, even if each of the key elements in the PPC is achieved, safe and effective *Shigella* vaccines may not be widely used where they are most needed.

This paradoxical situation is in part due to a potential “embarrassment of riches”: following the Decade of Vaccines [4] and the promulgation of the Global Vaccine Action Plan (GVAP) [5], there has been a steady progression in the development of numerous vaccine candidates directed at dozens of pathogens causing significant morbidity and mortality in low- and middle-income countries (LMICs). Multiple valuable public health interventions, not all of which are vaccines, are thus coming at a time of increasingly crowded immunization schedules and finite resources. The need to prioritize among these has prompted calls for critical and systematic assessments of the full public health value of individual vaccine candidates while still in development [6,7]. “Full public health value” refers to the full health, economic, and societal value of a vaccine as seen by a broad range of stakeholders, including local and global perspectives, and aims to articulate both its direct (individual) and indirect (population) effects. 

These assessments should build upon but transcend the traditional determinants of vaccine value in LMICs. Those determinants usually start with the demonstration that a target pathogen is responsible for “high” levels of hospitalization and mortality, as improved “child survival” was historically the major driver of vaccine development and introduction efforts for LMICs. A valuable vaccine may be expected to exhibit high (≥80%) efficacy against what is typically a clinical endpoint whose etiology is defined microbiologically or, increasingly, molecularly. An additional element of value is a cost-effectiveness ratio that sits within a somewhat arbitrary range of acceptability [8].

The above elements are fundamental to determining a vaccine’s potential value, but the different actors involved in the vaccine ecosystem may each have somewhat divergent views on their relative importance and have other considerations as well. For example, regulators value, above all, robust scientific and clinical data on vaccine safety and efficacy. Global and regional recommending bodies and foundations and trusts funding R&D for vaccines designed for use in LMICs will rely on these, but also must see a compelling risk-benefit profile at the population level, e.g., taking into account any herd protection and potential impact on antimicrobial-resistant pathogens.

The concerns of national immunization technical advisory groups (NITAGs) often go further, as they also put major emphasis on the ability—or not—for the specific vaccine under consideration to be feasibly delivered to target populations within their respective immunization schedules, and the incremental operational costs of doing this and accommodating the additional cold chain storage needs. They will also directly assess value relative to that of competing vaccines against other diseases on their public health agenda. Other country-level decision-makers may be concerned that demonstration of high efficacy against pathogen-specific endpoints may not readily translate into a correspondingly large—and visible—impact on the overall clinical syndrome, i.e., precisely what is actually perceived by clinicians and patients. They may also weigh vaccine value versus alternative interventions that may be easier or cheaper to implement. For example, malaria control includes interventions such as seasonal malaria chemoprevention (SMC) delivering oral drugs to young children monthly during peak malaria seasons. An injectable product with a similar or even higher efficacy may struggle to replace this if the challenges of needle and syringe delivery on a mass scale within a specific time window prove too great a burden relative to oral drug delivery. Where cost is high, malaria control programs may also consider the trade-off in terms of the number of diagnostic kits and treatments that could be procured for the same amount of funding. As the prolonged discussion over the malaria vaccine RTS,S has illustrated [9], an injectable product, even a vaccine, would have to demonstrate a very favorable profile across multiple domains to tip the scales in its favor.

Purchasers such as Gavi, the Vaccine Alliance (Gavi) may place more value on the high-level public health return and comparative cost-effectiveness of investments that can be used to attract sustainable contributions from donors. They may also ascribe special value to those vaccines that are already seen as a priority by Gavi-eligible LMICs, even if their public health benefit may be lower than others by some criteria.

Finally, vaccine developers see value in terms of the numbers of vaccine doses they can foreseeably sell at a price that makes sense within the context of their investment risk. Their perspective is often overlooked by public sector funders and policymakers, either through a limited understanding of sustainable business models or fear of association with the perceived ‘taint’ of profit. It is very difficult for developers to justify expensive product development, clinical studies, scale-up, and manufacture without confidence that a minimal return on investment can be realized.

## 2. Major Determinants of the Relative Value of Shigella Vaccines

While *Shigella* causes disease in all countries and age groups, current efforts to develop an effective *Shigella* vaccine are aimed at two high-risk groups in particular: children in LMICs and travelers/military from higher-income countries. As these target populations have quite different thresholds in terms of acceptable efficacy levels and other product attributes, for simplicity this discussion will focus on children, for whom there is good reason to believe that a highly effective vaccine against Shigella disease is indeed en route in the next several years.

The evidence begins with the proof-of-concept study cited earlier, a randomized, double-blind, placebo-controlled trial of a protein-polysaccharide conjugate vaccine targeted against the O-specific domain of the major *Shigella* lipopolysaccharides (LPS). It showed 70% efficacy against Shigellosis in 3–4-year-old Israeli children caused by one of the two major disease-causing serogroups, *S. sonnei*, [2]. While this study could not demonstrate statistically significant efficacy against disease in younger children or against *S. flexneri,* the other major disease-causing serogroup, several other researchers have built on these results to develop distinct, potentially more immunogenic multivalent formulations targeting the same O-sp domain of multiple serotypes. The most advanced parenteral candidates include synthetic conjugate [10] and bioconjugate formulations [11], and another using *Shigella* outer membrane vesicles as a platform [12]. A fourth promising candidate contains detoxified LPS co-formulated with highly conserved invasion plasmid proteins [13]. These vaccine candidates are all in or entering Phase II studies and/or being evaluated in controlled human infection models (CHIM), with the expectation that one or more phase III study could start soon after.

Public health officials and recommending bodies looking at the coming vaccine efficacy data will compare current Shigella mortality figures with those attributable to other potentially vaccine-preventable diseases since the adoption of a new *Shigella* vaccine would compete with them for limited resources. Thirty years ago, WHO estimated *Shigella* to be one of the top causes of infectious disease mortality in children, responsible for almost 600,000 deaths annually in <5-year-olds [14]. In contrast, current estimates put the number of child deaths at 28–64,000 or 1/10 that amount [15]. The difference in these figures is probably due to the overall decreasing levels of diarrheal disease stemming from water, sanitation, and hygiene (WASH) interventions, combined with better diagnostic tools and epidemiological studies.

For comparison, malaria deaths in children <5 are estimated at 260,000 annually, and the moderately effective RTS,S vaccine has just received a WHO/SAGE recommendation for widespread use in children at risk [16]. Some countries may prioritize the introduction of an HPV vaccine where there still are over 300,000 deaths worldwide annually; ~190,000 of these occur in LMICs [17], most of which still lack HPV immunization programs.

Other new or improved “competing” vaccines becoming available over the same 5–10-year time frame could conceivably prevent more deaths than a highly effective *Shigella* vaccine. Tuberculosis mortality in all ages, theoretically preventable by an improved vaccine [18], amounts to 1.4 million deaths annually [19]. Similarly, of the 128,000 rotavirus deaths that still occur annually [20], 90,000 are estimated to be essentially unpreventable even with increased coverage with the current moderately effective live oral vaccines (LORVs) [21]. A large portion of these might be prevented if LORVs were replaced by a theoretically more effective parenteral subunit formulation currently in Phase III studies [22]. Such a vaccine could also “compete” with *Shigella* vaccines for a place in the crowded immunization schedule.

Of course, there are other potential drivers of *Shigella* vaccine value. In addition to causing a significant but not overwhelming fraction of child mortality, an effective *Shigella* vaccine could prevent a major cause of dysentery, and molecular diagnostic techniques have recently revealed that *Shigella* represents a much greater cause of watery diarrhea than previously estimated [23]. One model estimated 111 million episodes annually in children <5 in 79 higher risk countries [24]. However, *Shigella* only represents one of several enteric pathogens causing diarrhea, and it is not clear whether a vaccine with only a partial impact on a clinical syndrome usually treated empirically (i.e., without an etiological diagnosis) would be considered a major priority. For country decision-makers, an alternative investment could be greater emphasis on non-pharmaceutical interventions to prevent diarrhea more broadly.

Two other aspects of *Shigella* should increase the perceived value of an effective vaccine. First, a *Shigella* vaccine would address an important cause of antibiotic use, as dysentery is one of the few diarrheal diseases for which antibiotic treatment is generally recommended. It would target a pathogen whose increasingly high levels of resistance to a variety of antibiotics make it a “serious threat” (according to the US Centers for Disease Control and Prevention) [25], and for which WHO deems a vaccine is “critically needed” [26].

Secondly, an effective vaccine might prevent a number of serious long-term sequelae associated with *Shigella* disease and perhaps even asymptomatic infection [27]. 2.1 million cases of moderate-severe stunting were attributed annually to *Shigella* in the same group of 79 lower- and middle-income countries [24]. This has been linked to increased infectious disease mortality due to other pathogens, perhaps associated with environmental enteric dysfunction, and the associated mortality has been estimated to account for an additional ~25% beyond that caused by acute *Shigella* disease alone. Because these additional effects might significantly increase the perceived value of an effective vaccine [28], a recent expert consultation was devoted in its entirety to examining the strength of the association between *Shigella* infection and growth faltering and the magnitude of the potential impact an effective *Shigella* vaccine could have on these long-term outcomes [29]. New impact model results are anticipated shortly.

With regard to economic value, the incremental cost-effectiveness ratio (ICER) of a *Shigella* vaccine delivered in infancy in WHO/AFRO countries was estimated a few years ago to be in the range of $1980 per disability-adjusted life year (DALY) averted [30] when stunting-related impacts were included. By way of comparison, the live oral rotavirus vaccines (LORV) used in GAVI countries were recently estimated to cost $500–$1500/DALY averted, and a parenteral rotavirus vaccine candidate was even lower at $300–$400 [31]. For RTS,S in African countries, the ICER was ~$90/DALY averted [32], while for pneumococcal conjugates in African countries, it was $118 [33]. Though these analyses are not directly comparable due to differences in coverage and pricing assumptions and methodology, they provide some indication of a *Shigella* vaccine’s relative cost-effectiveness in those regions. They also suggest that the introduction of *Shigella* vaccines should be geographically more targeted than that for some of the other pathogens, perhaps even at a subnational level [34]. However, the economics and administrative challenges of subnational targeting are not straightforward and might require further analysis.

## 3. Further Efforts to Better Understand the Value of Shigella Vaccines

It is possible that the revised estimates of the overall *Shigella* disease burden and its association with stunting, currently under development, could significantly change its potential impact and cost-effectiveness relative to other vaccines [29]. It will be critical to complement those modelized estimates with a direct demonstration of the value of a *Shigella* vaccine via efficacy and effectiveness studies against those clinical outcomes considered by country stakeholders to be of greatest interest. For example, what do they consider to be the relative importance of demonstrating vaccine impact on *Shigella*-specific diarrhea versus against all-cause moderate-to-severe diarrhea versus growth faltering and stunting?

To determine the answer to this and other questions will require another activity that is often overlooked, namely a systematic consultation with LMIC stakeholders to explore which potential properties of a *Shigella* vaccine they consider to be the key drivers of value for its adoption and introduction. After all, as the resources needed to acquire and introduce these vaccines largely rest with countries, even where Gavi initially co-finances vaccine purchase, the ultimately decisive perspective is that of those who would adopt and integrate *Shigella* vaccines into their respective immunization programs.

To have sufficient validity, these consultations should comprise multi-country in-depth analyses that involve national opinion leaders, policymakers, and finance officials, as well as selected health care providers. These could include structured or semi-structured interviews, open-ended or discrete choice surveys, formal and informal consultations of many kinds, and successful examples exist for other pathogens [35,36]. Such analyses could also be useful to better understand what LMIC stakeholders consider to be the optimal schedule for a *Shigella* vaccine. With the highest risk window for shigellosis in the second year of life, it is currently envisioned that vaccination could begin at 6 months of age and include a 2nd dose 3 or 6 months later [3]. However, in many countries, especially in Africa, there is no EPI visit at 6 months (though RTS,S may change that). Furthermore, at 9 and 12 months of age, multiple other vaccines may already be given. Consultations with the individuals involved in immunization program operations would thus help us understand whether the addition of a *Shigella* vaccine would create significant implementation challenges and how they might be mitigated.

A related question is whether the delivery of a *Shigella* vaccine in combination with another vaccine would significantly change the picture. This is purely hypothetical at this point because the development, licensure, and manufacture of a new combination vaccine comes with its own unique set of challenges and risks. Nevertheless, combining *Shigella* with an existing vaccine could eliminate the need for additional injections and reduce delivery costs, obviate additional burden on the cold-chain, and, if combined with a vaccine directed against another diarrheal pathogen, should result in a greater impact at the syndromic level. This might be particularly attractive since, outside of a research study setting, individual cases of diarrhea in LMICs are rarely attributed to a specific etiology, and prevention of *Shigella* disease by itself may thus not be a compelling feature. Finally, querying national stakeholders could help us understand whether the perceived value of *Shigella* vaccines would be increased if they were targeted at a subnational level. It would also help us understand what kinds of operational/implementation research are needed to assist with decision making.

## 4. Conclusions

The development of highly effective and safe *Shigella* vaccines would be a major accomplishment, but these vaccines will not sell themselves in the context of a crowded immunization schedule and many other competing interventions. There is still work to be conducted to determine if there is a compelling public health value proposition that can help stimulate the political will and grassroots advocacy often necessary for vaccine recommendations, purchase, and adoption. The oft-underestimated political will may be most efficiently stimulated when there is a clear, compelling, and succinct description of why a vaccine is valuable and for which priority target populations, rather than a list of all of its possible benefits (even if accurate). Successful models may include “preventing one of the biggest causes of cancer in women” as in the case of HPV vaccines, or “eliminating the leading cause of epidemic meningitis in Africa” in the case of meningococcal A conjugates. Such “value propositions” take into account the reality that many decision-makers are policymakers and finance officials, not disease area experts.

There are multiple potential determinants of *Shigella* vaccine value (Table 1). We suggest that articulation of a succinct and compelling value proposition for *Shigella* vaccines will be helpful for their prioritization and adoption to achieve the public health goals as originally envisioned.

## Figures and Tables

**Table 1 vaccines-10-00282-t001:** Some potential determinants of *Shigella* vaccine value in the context of competing interventions.

Potential Value Driver	Current Relevant Information	Comments
Mortality in target population	28,000–60,000 deaths <5 annually [15]	Significant, but lower than several other potential vaccines for their respective target populations
Morbidity	111 million diarrhea episodes/yr in <5; 2.1 million cases of moderate-severe stunting [24]. Significant morbidity burden in other key populations (e.g., older children, travelers, and military).	Child mortality, not morbidity, has been the traditional driver of vaccine prioritization, but this may change.
Cost-effectiveness	$1980 per disability-adjusted life year averted (for WHO/AFRO countries) [30]	Higher than for a number of other new vaccine candidates, but likely will decrease in new estimates.
Antibiotic Resistance	*Shigella* targeted by WHO and US CDC as an important potential vaccine target due to increasing levels of resistance [25,26].	Antimicrobial resistance has not yet emerged as a major driver of vaccine prioritization.
Integration into existing EPI schedule	2 doses to be delivered at 6–12 months of age [3]	Depending on the precise timing, there may not be a visit in many countries to accommodate both doses and/or several other vaccines may be given at the same time. Latter concern could be alleviated if *Shigella* is combined with another vaccine.
Prospect of a healthy, competitive, and sustainable vaccine market	There are multiple vaccine developers with promising candidates, but unclear whether current expected pricing and demand will be considered to allow a sufficient return on investment	Obviously subjective, but the more clarity on the likely magnitude of demand and pricing, the easier to evaluate the prospect of sustained interest from developers. Demand from other populations in high-income countries (e.g., travelers and military) can also influence commercial sustainability.
What LMIC stakeholders think	Systematic consultation with stakeholders [35,36]	Ultimate deciders of vaccine prioritization and introduction

## Data Availability

Not applicable.

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
