# Peer review of "What Drives the Value of a Shigella Vaccine?"

_vaccines, 2022, doi:10.3390/vaccines10020282_

Round 1

Reviewer 1 Report

I appreciate the opportunity to review this manuscript and would like to tell that I appreciate it very much. I enjoyed reviewing it. Congratulations.

Considering that the manuscript is for a special issue of Frontiers in Shigella Vaccine Development, I found that it meets the scope of the issue and I rated the overall merit of the manuscript High. I am pleased to recommend to accept the manuscript in the present form.

Author Response

Thanks for nice comments.

Reviewer 2 Report

This is valuable scientific contribution about the development of a vaccine (Shigella vaccine) against a global disease causing high morbidity and mortality, especially in children under five. The authors could improve this interesting presentation by mentioning that they demonstrate an important question based on a specific development of a Shigella vaccine. To make the reader aware that this paper deals with two main questions, a general broader platform on the prevention of morbidity and mortality, and the second of specific importance on risk/benefit, cost/benefit, financial support organization of Shigella vaccine in developing countries.

The main question answered by the research is the question of the priority of a Shigella vaccine in humans. The introduction of a Shigella vaccine to the market must overcome numerous hurdles. The driving arguments for and against the introduction of a Shigalla vaccine are laid out. The paper argues for a concise value proposition to move current developments along.

A Shigella vaccine is very important, has been set as a high priority by the WHO with a perspective of availability in ten years. It has to be considered though there are other important gastrointestinal pathogens which have been known for decades with an even higher priority for which we still do not have a vaccine e.g. E.coli as well as viral and parasitic diseases. The paper deals with the driving value factors of such a vaccine and is in this regard valuable to the field, albeit may be of lesser importance to the audience of scientific journals. There is no platform on this topic as far as I know at the level of journals between “Journal of Experimental Medicine” and “Pathogens”. It is meant as a question to the editors whether they would want to start such a topic.

It adds numerous questions regarding socio-economical, financial and political considerations, but does not discuss the present scientific and technical situation regarding Shigella research. It leaves it to the WHO and other cited reports. The authors should add a brief overview on the scientific progress and not only rely on the cited references. On the other hand, a discussion on how to successfully launch a vaccine is necessary and requires a coordinated effort between the scientific community and public health decision makers of the respective countries. Papers like this can be useful to bridge this gap as it is written in a way that appeals to a broad audience. Now more than ever researchers need to publicly communicate the value of their findings. This paper is therefore an important contribution

A significant improvement would be the integration of the current state of research on Shigella. In its current state the interest of “Vaccines” audience may be moderate.

The conclusions are consistent with the evidence and answer the question posed by the authors. The references are well researched and appropriate.

The authors have included only one table with convoluted information. It would be favorable subdividing it into further tables or figures. 

Author Response

  1. a) “It adds numerous questions regarding socio-economical, financial and political considerations, but does not discuss the present scientific and technical situation regarding Shigella It leaves it to the WHO and other cited reports… The authors should add a brief overview on the scientific progress and not only rely on the cited references. A significant improvement would be the integration of the current state of research on Shigella.”

We thank this and another reviewer for this helpful suggestion, and we have now included information about scientific progress and the current vaccine candidates at the very beginning of section 1, lines 22-25:

Proof of concept was established more than a decade ago that a vaccine effective in adults against the major diarrheal disease pathogen Shigella [1] could also prevent the disease in children [2]. Based on those studies and the multitude of Shigella vaccine candidates in Phase II studies, it is easy to be optimistic…

and more extensively in section 2, lines 98-119.  (new text in italics)

As these target populations have quite different thresholds in terms of acceptable efficacy levels and other product attributes, for simplicity this discussion will focus on children, for which there is good reason to believe that a highly effective vaccine against Shigella disease is indeed en route in the next several years.

For example, the proof-of-concept study cited earlier was a randomized double-blind, placebo-controlled trial of a protein-polysaccharide conjugate vaccine targeted against the O-specific domain of the major Shigella lipopolysaccharides. It showed 70% efficacy against Shigellosis in 3-4 year old Israeli children caused by one of the two major disease-causing serogroups, S. sonnei, [2]. While this study could not demonstrate statistically significant efficacy against disease in younger children or against the other major disease-causing serogroup, S. flexneri, several other researchers have built on these results to develop distinct, potentially more immunogenic multivalent formulations targetting the same O-sp domain of multiple serotypes. The most advanced parenteral candidates include synthetic conjugate [10] and bioconjugate formulations [11], and another using Shigella outer membrance vesicles as a platform [12]. A fourth promising candidate contains detoxified LPS co-formulated with highly conserved invasion plasmid proteins [13].  These vaccine candidates are all in or entering Phase II studies and/or being evaluated in controlled human infection models (CHIM), with the expectation that one or more phase III study could start soon after.

Public health officials and recommending bodies looking at the coming vaccine efficacy data will compare current Shigella mortality figures with those attributable to other potentially vaccine-preventable diseases…. 

  1. b) “The authors have included only one table with convoluted information. It would be favorable subdividing it into further tables or figures.”

We thank the reviewer for the feedback.  Upon review of the type-set version, we have rephrased perhaps the most “convoluted” row of the table (involving commercial sustainability) to improve readability (revised wording in italics).  We also believe that having the type-set table on a single page will obviate the need to divide it into separate tables.

Prospect of a healthy,  competitive and sustainable vaccine market

There are multiple vaccine developers with promising candidates, but unclear whether current expected pricing and demand will be considered to allow a sufficient return on investment

Obviously subjective, but the more clarity on likely magnitude of demand and pricing, the easier to evaluate the prospect of sustained interest from developers.  Demand from other populations in high income countries (e.g., travelers and military) can also influence commercial sustainability.

Reviewer 3 Report

Authors use existing knowledge in factors used to determine the public health significance of implementing newly developed vaccinations and apply that knowledge to potential Shigella vaccine. The commentary is well-written and easy to comprehend. They authors describe both positive and negative sides of the argument and articulate their opinion.

Author Response

Thanks for reviewing.

Reviewer 4 Report

The manuscript entitled ‘What Drives the Value of a Shigella Vaccine?’ by Hausdorff et. al. summarizes the current issues faced by Shigella vaccines for development and licensure. The authors discuss value addition points for Shigella vaccine and highlight the bottlenecks. The article is well written and makes lot of sense not only for Shigella vaccines but many other vaccines which are still in development process. The article is nicely structured and I have only couple of more points to add-

1- It will be good idea to include some information about current vaccine candidates for Shigella so that readers in this particular field will get attracted to this article. I believe that current draft is more focused on science policy that science itself. May be authors are aiming that but adding some information about the Shigella vaccine candidates will make article available to a broader audience.

2- One more piece of information - that authors can think about adding - is cost of these vaccine candidates and feasibility of their distribution. For middle and low income countries, cost is major bottleneck and lack of vaccine delivery services makes this more complicated. The vaccine candidates that don’t require cold chain will be more successful in these settings and will cost less. This point can considered to be added in the manuscript.

Author Response

1- It will be good idea to include some information about current vaccine candidates for Shigella so that readers in this particular field will get attracted to this article…

We thank this and another reviewer for this helpful suggestion, and we have now included information about scientific progress and the current vaccine candidates at the very beginning of section 1, lines 22-25:

Proof of concept was established more than a decade ago that a vaccine effective in adults against the major diarrheal disease pathogen Shigella [1] could also prevent the disease in children [2]. Based on those studies and the multitude of Shigella vaccine candidates in Phase II studies, it is easy to be optimistic…

and more extensively in section 2, lines 98-119.  (new text in italics)

As these target populations have quite different thresholds in terms of acceptable efficacy levels and other product attributes, for simplicity this discussion will focus on children, for which there is good reason to believe that a highly effective vaccine against Shigella disease is indeed en route in the next several years.

For example, the proof-of-concept study cited earlier was a randomized double-blind, placebo-controlled trial of a protein-polysaccharide conjugate vaccine targeted against the O-specific domain of the major Shigella lipopolysaccharides. It showed 70% efficacy against Shigellosis in 3-4 year old Israeli children caused by one of the two major disease-causing serogroups, S. sonnei, [2]. While this study could not demonstrate statistically significant efficacy against disease in younger children or against the other major disease-causing serogroup, S. flexneri, several other researchers have built on these results to develop distinct, potentially more immunogenic multivalent formulations targetting the same O-sp domain of multiple serotypes. The most advanced parenteral candidates include synthetic conjugate [10] and bioconjugate formulations [11], and another using Shigella outer membrane vesicles as a platform [12]. A fourth promising candidate contains detoxified LPS co-formulated with highly conserved invasion plasmid proteins [13].  These vaccine candidates are all in or entering Phase II studies and/or being evaluated in controlled human infection models (CHIM), with the expectation that one or more phase III study could start soon after.

Public health officials and recommending bodies looking at the coming vaccine efficacy data will compare current Shigella mortality figures with those attributable to other potentially vaccine-preventable diseases…. 

2- One more piece of information - that authors can think about adding - is cost of these vaccine candidates and feasibility of their distribution. For middle and low income countries, cost is major bottleneck and lack of vaccine delivery services makes this more complicated. The vaccine candidates that don’t require cold chain will be more successful in these settings and will cost less. This point can considered to be added in the manuscript.

We appreciate this suggestion and have now made more explicit reference to the cold chain need, both in the first section (lines 64-65):

…and the incremental operational costs of doing so and accommodating the additional cold chain storage needs.  They will also directly…

and the third section (lines 236-237)

…Nevertheless, combining Shigella with an existing vaccine could eliminate the need for additional injections and reduce delivery costs, obviate additional burden on the cold-chain and, if combined…